# Transcriptional Expression of SLC2A3 and SDHA Predicts the Risk of Local Tumor Recurrence in Patients with Head and Neck Squamous Cell Carcinomas Treated Primarily with Radiotherapy or Chemoradiotherapy

**DOI:** 10.3390/ijms26062451

**Published:** 2025-03-09

**Authors:** Mercedes Camacho, Silvia Bagué, Cristina Valero, Anna Holgado, Laura López-Vilaró, Ximena Terra, Francesc-Xavier Avilés-Jurado, Xavier León

**Affiliations:** 1Genomics of Complex Diseases, Research Institute Hospital Sant Pau, IIB Sant Pau, 08041 Barcelona, Spain; mcamacho@santpau.cat; 2Pathology Department, Hospital de la Santa Creu i Sant Pau, Universitat Autònoma de Barcelona, 08041 Barcelona, Spain; sbague@santpau.cat (S.B.); llopezv@santpau.cat (L.L.-V.); 3Otorhinolaryngology Department, Hospital de la Santa Creu i Sant Pau, Universitat Autònoma de Barcelona, 08041 Barcelona, Spain; cvalero@santpau.cat (C.V.); aholgado@santpau.cat (A.H.); 4MoBioFood Research Group, Biochemistry and Biotechnology Department, Universitat Rovira i Virgili, Campus Sescelades, 43007 Tarragona, Spain; ximena.terra@urv.cat; 5Head Neck tumors Unit, Hospital Clínic Barcelona, Universitat de Barcelona, IDIBAPS, 08036 Barcelona, Spain; faviles@clinic.cat; 6Centro de Investigación Biomédica en Red de Bioingeniería, Biomateriales y Nanomedicina (CIBER-BBN), 28029 Madrid, Spain

**Keywords:** SLC2A3, GLUT3, SLC16A, SUCNR1, SDHA, biomarker, Warburg effect, head and neck squamous cell carcinoma

## Abstract

Reprogramming of metabolic pathways is crucial to guarantee the bioenergetic and biosynthetic demands of rapidly proliferating cancer cells and might be related to treatment resistance. We have previously demonstrated the deregulation of the succinate pathway in head and neck squamous cell carcinoma (HNSCC) and its potential as a diagnostic and prognostic marker. Now we aim to identify biomarkers of resistance to radiotherapy (RT) by analyzing the expression of genes related to the succinate pathway and nutrient flux across the cell membrane. We determined the transcriptional expression of succinate receptor 1 (SUCNR1), succinate dehydrogenase A (SDHA), and the solute carrier (SLC) superfamily transporters responsible for the influx or efflux of a wide variety of nutrients (SLC2A3 and SLC16A3) in tumoral tissue from 120 HNSCC patients treated with RT or chemoradiotherapy (CRT). Our results indicated that the transcriptional expression of the glucose transporter SLC2A3 together with SDHA had the best predictive capacity for local response after treatment with RT or CRT. High SLC2A3 and SDHA expression predicted poor outcomes after RT or CRT, with these patients having a 4.2 times higher risk of local recurrence compared to the rest of the patients. These results might indicate that tumors that shifted toward a higher glucose influx and a higher oxidation of succinate via mitochondrial complex II present an ideal environment for radioresistance development. Patients with a high transcriptional expression of both SLC2A3 and SDHA had a significantly higher risk of local recurrence after treatment with RT or CRT.

## 1. Introduction

Radiotherapy (RT) is widely used in patients with head and neck squamous cell carcinoma (HNSCC), either alone or in multimodality therapeutic strategies including surgery and chemotherapy. Currently, there are no genomic or molecular markers other than p16 or human papillomavirus (HPV) status that can effectively predict the outcome of RT in patients with HNSCC. Accordingly, the identification of molecular markers that predict response to RT would be an important milestone in head and neck oncology and would help move toward the development of personalized treatments to maximize survival and minimize morbidity [1].

Aberrant glucose metabolism has been recognized as a common feature in cancer for nearly a century [2]. This phenomenon, known as the Warburg effect, consists of a significantly elevated rate of glucose consumption and lactate excretion that is largely insensitive to oxygen availability [3]. There is significant evidence to demonstrate the metabolic reprogramming of tumor cells, as an addition to excessive uptake and metabolism of key nutrients to support rapid proliferation and invasion capacities. The transport of key nutrients for cancer metabolism, such as glucose and amino acids, is of particular interest for their roles in tumor progression and metastasis. Solute carrier (SLC) superfamily transporters are responsible for the influx or efflux of a wide variety of nutrients that are needed for the cells to function. To meet the increased demand for nutrients and energy, SLC transporters are frequently deregulated in cancer cells. For example, elevated SLC2A1 (GLUT1) and SLC2A3 (GLUT3), which facilitate the transport of glucose across the plasmatic membrane, have been associated with increased cancer metabolism [4]. Specifically, the role of glucose transporters in oral squamous cell carcinoma has been recently reviewed by Botha et al. [5]. The authors concluded that SLC2A1 and SLC2A3 have a role in the pathophysiology of oral squamous cell carcinoma and represent valuable biomarkers in assessing diagnosis and prognosis. SLC16A family lactate transporters have also been reported to be deregulated, in part to maintain intracellular pH for continued growth with increased lactate production [6].

The tricarboxylic acid (TCA) cycle is strategically situated at the center of cellular metabolism, where it serves as an anabolic hub for the synthesis of macromolecules such as fatty acids, cholesterol, and amino acids that are crucial to support rapidly growing tumors. The TCA cycle must be constantly replenished by carbon atoms, a process named anaplerosis [7]. The current dogma in biochemistry is that the main anaplerotic substrates are pyruvate (from glucose), which is converted into oxalacetate, and glutamate (from glutamine), which is metabolized to succinate, with lesser contributions from precursors of propionyl-CoA, such as odd-chain fatty acids, amino acids, and C5 ketone bodies that also feed into succinate [8].

Our group has previously demonstrated that the circulating succinate levels are elevated in HNSCC, identifying this oncometabolite as a potentially valuable non-invasive biomarker for HNSCC diagnosis [9]. Furthermore, we demonstrated an important role of the succinate-related pathway in tumor development and response to treatment in patients with HNSCC, where high succinate receptor 1 (SUCNR1) together with high succinate dehydrogenase A (SDHA) expression predicts poor locoregional disease-free survival in a sub-group of patients treated with RT or chemoradiotherapy (CRT) [9].

The close relationship between the transporters of anaplerotic substrates and the succinate pathway in the TCA cycle as an anabolic hub for the synthesis of macromolecules to support rapidly growing tumors made us hypothesize that the potential of SUCNR1/SDHA as prognostic biomarkers could be enhanced including in the model transporters of key nutrients that are crucial for cancer metabolism.

The present study aims to evaluate the prognostic potential of the transcriptional expression of genes encoding solute carrier transporters (SLC2A3 and SLC16A3) and to develop a predictive model that integrates these genes with others associated with the succinate pathway (SUCNR1 and SDHA), thereby enhancing prognostic accuracy. The analysis was conducted using a cohort of patients with HNSCC treated with RT or CRT.

## 2. Results

We analyzed 120 patients with squamous cell carcinomas located in the oral cavity, oropharynx, hypopharynx, or larynx treated primarily with RT or CRT. During the follow-up period, 39 patients (32.5%) had a local recurrence of the tumor, 14 (11.7%) had a regional recurrence, and 15 (12.5%) had distant metastases. We defined local recurrence as the persistence or recurrence of carcinoma at the primary tumor site following completion of radiotherapy or chemoradiotherapy.

### 2.1. Transcriptional Expression of SUCNR1, SLC2A3, SLC16A3, and SDHA

When analyzing the transcriptional expression according to clinical variables, we observed significant differences according to sex in the expression of SLC2A3. SLC2A3 expression was higher in female patients. There were no significant differences in the expression of the genes analyzed according to the history of toxics consumption or local extension (cT) or regional extension (cN) of the tumor. In the case of patients with oropharyngeal carcinomas, SUCNR1 expression was higher in patients with HPV-positive tumors.

There were significant differences in the SLC2A3 transcriptional expression according to the local tumor control after RT or CRT. Patients with local recurrence had significantly higher SLC2A3 expression values than patients who had local control after treatment (*p* = 0.043). Patients with a local recurrence tended to have higher SDHA expression, but without reaching statistical significance (*p* = 0.085). Appendix A of the Appendix A shows the distribution of SLC2A3 expression according to local control. Appendix A of the Appendix A shows the median transcriptional expression values of the genes analyzed according to the different variables studied.

### 2.2. Results of the Recursive Partitioning Analysis (RPA)

We individually evaluated with an RPA the relationship between the expression values of each gene and local disease control after treatment. We found a significant relationship between the high expression of SLC2A3, SLC16A3, and SDHA and a higher rate of local recurrence. No relationship was found between SUCNR1 expression and local disease control. Table 1 shows the distribution of patients according to the cut-off values obtained with the RPA, as well as the 5-year local recurrence-free survival for each of the categories obtained with these cut-off values. Appendix A of the Appendix A shows the local recurrence-free survival curves according to the transcriptional expression categories of SLC2A3, SLC16A3, and SDHA obtained with the RPA.

When we jointly analyzed the transcriptional expression values of the three genes that were related to local disease control, the RPA model classified patients into three categories, with the first level of classification based on SLC2A3 expression and the second level of classification for patients with high SLC2A3 expression based on SDHA expression. A classification tree with three terminal nodes was obtained (Figure 1): patients with low SLC2A3 expression (n = 64, percentage of local recurrence 21.9%), patients with high SLC2A3 and low SDHA expression (n = 21, percentage of local recurrence 14.3%), and patients with high SLC2A3 and high SDHA expression (n = 35, percentage of local recurrence 62.9%). We then proceeded to group the two terminal nodes with lower local recurrence rates, classifying patients into two groups: Group 1, patients with low SLC2A3 expression and patients with high SLC2A3 and low SDHA expression (n = 85, local recurrence rate 20.0%), and Group 2, patients with high SLC2A3 and high SDHA expression (n = 35, local recurrence rate 62.9%).

Appendix A presents the distribution of patients according to the SLC2A3-SDHA expression category in relation to clinical variables. Significant differences were observed only in the distribution of patients based on the extension of the primary tumor. Specifically, the frequency of patients with early-stage tumors (cT1-2) was higher in Group 1 compared to Group 2 (78.6% vs. 21.4%, *p* = 0.041).

### 2.3. Survival According to the SLC2A3-SDHA Expression Category

Figure 2 shows the local recurrence-free survival according to the transcriptional SLC2A3-SDHA group. Five-year local recurrence-free survival for Group 1 was 79.1% (95% CI:70.3–87.9%), and for Group 2, it was 35.1% (95% CI:18.7–51.5%) (*p* = 0.0001).

The observed differences in local disease control were independent of the treatment modality. For patients treated with RT (n = 54), 5-year local recurrence-free survival for Group 1 (n = 42) was 87.7% (95% CI: 77.7–97.7%), and for Group 2 (n = 12), it was 41.7% (95% CI: 13.9–69.5%) (*p* = 0.0001). For patients treated with CRT (n = 66), 5-year local recurrence-free survival for Group 1 (n = 43) was 70.6% (95% CI: 56.5–84.7%), and for Group 2 (n = 23), it was 31.4% (95% CI: 11.0–51.8%) (*p* = 0.001). The local recurrence-free survival curves, stratified by the SLC2A3-SDHA transcriptional group and treatment type, are shown in Appendix A of the Appendix A.

Similarly, we observed differences in local disease control regardless of the local extension of the primary tumor. For patients with early tumors (cT1-T2, n = 70), 5-year local recurrence-free survival for Group 1 (n = 55) was 85.0% (95% CI: 75.4–94.6%), and for Group 2 (n = 15), it was 53.3% (95% CI: 28.0–78.6%) (*p* = 0.005). In patients with locally advanced tumors (cT3-T4, n = 50), 5-year local recurrence-free survival for Group 1 (n = 30) was 68.3% (95% CI: 51.1–85.5%), and for Group 2 (n = 20), it was 21.4% (95% CI: 2.0–40.8%) (*p* = 0.001). Appendix A in the Appendix A shows the local recurrence-free survival curves according to the SLC2A3-SDHA transcriptional group depending on the local tumor extension category.

### 2.4. Multivariable Analysis

Table 2 shows the result of a multivariable analysis in which local recurrence-free survival was considered as the dependent variable. The only variable that was significantly associated with local control was the category of SLC2A3-SDHA expression. Relative to patients in Group 1, patients in Group 2 (high SLC2A3 and high SDHA expression) had a 4.24-fold increased risk of local recurrence (95% CI: 2.07–8.69, *p* = 0.0001).

When analyzing patients with oropharyngeal carcinomas according to HPV status, we observed that in HPV-negative tumors, the advantage in local recurrence-free survival was maintained for patients in Group 1 (5-year local recurrence-free survival for patients in Group 1 66.9% versus 12.9% for patients in Group 2), although the differences did not reach statistical significance (*p* = 0.123). For patients with HPV-positive tumors, the 5-year local recurrence-free survival for patients in Group 1 (n = 9) was 88.9%, and for patients in Group 2 (n = 3), it was 100%.

### 2.5. Local Recurrence-Free Survival According to SLC2A3 and SDHA Transcriptional Expression

Table 3 shows the 5-year local recurrence-free survival rates according to whether patients had high or low expression of SLC2A3 and SDHA according to the cut-off points obtained in the individual analysis of each gene. A significant decrease in local recurrence-free survival was observed only in the combination of elevated SLC2A3 and elevated SDHA expression. Notably, for patients with low SLC2A3 expression (n = 64), no differences in 5-year local recurrence-free survival were seen according to the SDHA expression category (75.8% for patients with low SDHA expression vs. 78.6% for those with high SDHA expression, *p* = 0.826). Appendix A in the Appendix A shows the local recurrence-free survival of patients according to whether they had high or low SLC2A3 and SDHA expression.

### 2.6. Regional Recurrence and Distant Metastasis-Free Survival According to SLC2A3 and SDHA Transcriptional Expression

Regional recurrence-free survival for Group 1 was significantly higher than for Group 2 (5-year regional recurrence-free survival: 93.9%, 95% CI: 88.8–99.0% versus 71.5%, 95% CI: 55.0–88.0%, *p* = 0.001). Distant metastasis-free survival for Group 1 was also higher than Group 2, although in this case, the differences did not reach statistical significance (5-year distant metastasis-free survival: 88.7%, 95% CI: 81.6–95.8% versus 80.3%, 95% CI: 66.0–94.6%, *p* = 0.249). Group 1 patients had significantly better disease-specific survival. Five-year disease-specific survival for Group 1 was 75.6% (95% CI: 65.8–85.4%), and for Group 2, it was 51.2% (95% CI: 33.2–69.2%) (*p* = 0.005).

### 2.7. Results of the External Validation Study with the TCGA Data

Significant differences in SLC2A3 transcriptional expression were observed among patients in the TCGA cohort based on tumor status (*p* = 0.044). Patients who were tumor-free at the last contact or time of death (n = 340, 70.8%) exhibited significantly lower SLC2A3 expression compared to those with active tumors (n = 140, 29.2%). The distribution of SLC2A3 expression by tumor status is presented in Appendix A of the Appendix A. In contrast, SDHA expression did not differ significantly based on tumor status (*p* = 0.144).

When analyzing the transcriptional expression of SLC2A3 and SDHA with an RPA considering tumor status as the dependent variable, we obtained a classification tree with three terminal nodes, with SLC2A3 expression as the primary classifier and SDHA expression further stratifying patients with low SLC2A3 levels. Among patients with high SLC2A3 expression (n = 55), 49.1% had an active tumor at the last contact or time of death. In contrast, the proportion of patients with active tumors was 32.3% among those with low SLC2A3 and high SDHA expression (n = 189) and 22.3% among those with low SLC2A3 and low SDHA expression (n = 229). These differences were statistically significant (*p* = 0.0001). The classification and regression tree based on SLC2A3 and SDHA expression are presented in Appendix A of the Appendix A.

## 3. Discussion

According to our results, the joint assessment of SLC2A3 and SDHA transcriptional expression allowed us to define a group of patients with an elevated risk of local tumor recurrence after treatment with RT or CRT. Patients with elevated expression of both SLC2A3 and SDHA had a 4.24-fold increased risk of local recurrence relative to all other patients. The predictive capacity of SLC2A3-SDHA expression was consistent, regardless of the treatment modality (RT or CRT) or the local extension of the tumor (cT1-T2 or cT3-T4).

Elevated expression of glucose transporters is associated with decreased survival in most cancer models, including HNSCCs [4,10]. In a study of patients with surgically treated oral cavity carcinomas, Ayala et al. [11] found that patients with high immunohistochemical expression of SLC2A3 had a significantly increased risk of recurrence and decreased survival in both uni- and multivariable analyses. In another study carried out in patients with oral cavity carcinomas treated with surgery, Estilo et al. [12] found a significant association between elevated SLC2A3 transcriptional expression and depth of invasion, pathologic staging, and risk of recurrence. Similarly, in patients with laryngeal carcinomas treated with surgery and/or radiotherapy, Baer et al. [13] found a significant association between the immunohistochemical expression of SLC2A3 and survival. In patients with advanced laryngeal carcinomas treated with surgery, Starska et al. [14] found an increase in the transcriptional and immunohistochemical expression of SLC2A3 in tumor tissue relative to adjacent normal laryngeal tissue. Tumors with elevated SLC2A3 expression tended to have worse survival, but the differences did not reach statistical significance. In addition, SLC2A3 transcriptional expression appears in genetic signatures associated with prognosis and response to treatment in patients with HNSCC [15,16].

Some authors have found a relationship between SLC2A1 expression and response to RT in HNSCC patients. Kunkel et al. found that HNSCC with a high immunohistochemical expression of SLC2A1 had increased resistance to RT [17]. Chen et al. observed a significant decrease in disease-free survival and disease-specific survival in patients with p16-negative oropharyngeal or hypopharyngeal carcinomas treated with RT or CRT and a high immunohistochemical expression of SLC2A1 [18]. Nonetheless, to our knowledge, there are no studies that have analyzed the relationship between SLC2A3 expression and response to RT.

SDHA is a mitochondrial TCA cycle enzyme that converts succinate to fumarate. Mutations in the gene encoding SDHA that result in altered SDHA function have been associated with the appearance of tumors such as paragangliomas and pheochromocytomas, gastrointestinal stromal tumors, renal cell carcinomas, and pituitary adenomas [19]. However, there is no evidence that SDHA itself is mutated or dysfunctional in HNSCCs.

In a previous study from our group in a cohort of 41 patients with HNSCC independent of the cohort of the present study, we were able to demonstrate that SDHA with elevated transcriptional expression was associated with the loco-regional control of the disease in patients treated with RT or CRT [9]. In this study, we performed an external validation of the prognostic capacity of SDHA expression in patients treated with RT or CRT.

The number of studies that have analyzed the involvement of SDHA in the process of carcinogenesis is limited. Chattopadhyay et al. [20] found that high expression of SDHA was associated with the risk of metastatic spread and poor clinical outcome in patients with uveal melanoma, and Olszewski et al. found inhibition of tumor growth in a patient-derived xenograft model of SDHA-deficient tumors [21]. On the other hand, SDHA has been described as a tumor suppressor because it reduces the accumulation and secretion of succinate, considered an oncometabolite [22]. Li et al. found that SDHA was frequently downregulated in hepatocellular carcinoma tissues and that this downregulation was associated with poor prognosis [23]. By analyzing the data included in The Cancer Genome Atlas (TCGA), it can be observed that the prognostic capacity of SDHA transcriptional expression depends on the tumor type [24]. High expression of SDHA was associated with significantly reduced overall survival in breast cancer patients, whereas it was associated with increased survival in renal cancer patients. For patients with HNSCC, high SDHA expression was associated with a decrease in overall survival, but the differences did not reach statistical significance (*p* = 0.16). It should be considered that patients included in TCGA are mainly patients with oral cavity carcinomas and a large majority of patients treated with surgery, which significantly differs from the patients analyzed in our study.

Dysregulation in SDH may exert effects on many metabolic pathways [25]. Recently, Schöpf et al. [26] studied the rewiring of metabolism in prostate cancer. Their results reveal a shift toward higher oxidation of succinate, which is associated with deleterious mutations in mitochondrial Complex I genes, and a rewired expression of mitochondrial metabolic enzymes. Their study on oxidative phosphorylation capacities in prostate cancer tissues uncovered increased oxidation of succinate via Complex II as compensation for a decreased capacity to oxidize substrates via Complex I. In our cohort of HNSCC patients, the increased SDHA expression might also be associated with an increased oxidative phosphorylation capacity which might confer radiation resistance. Nonetheless, SDH enzymes are characterized by fine regulatory mechanisms including regulation of mRNA expression, post-translational modification, and endogenous inhibition [27]. Further studies are needed to investigate the role of this enzyme in both the carcinogenesis and progression of HNSCC.

The nutrient supply in tumor cells may originate from cellular metabolism or may be imported from the external microenvironment across the plasma membrane via transporters, fueling TCA for the synthesis of metabolic precursors, and OXPHOS for ATP production in mitochondria. Therefore, on one hand, a high expression of SLC2A3 might enhance aerobic glycolysis and the production of lactate, and on the other hand, an increased expression of SDH might favor (I) the conversion of succinate to fumarate as part of the tricarboxylic acid cycle and (II) the oxidative phosphorylation, delivering reducing equivalents into the electron transport chain via FADH2 and finally producing ATP. This shift seems an ideal environment for cancer development because it allows sparing of bio precursors for other needs, otherwise used for NADH production, while high succinate oxidation is maintained for efficient ATP production using FADH2 as an electron donor. This hypothesis agrees with the fact that in our results, when SLC2A3 expression is low, SDHA (high or low expression levels) does not affect the outcomes. This result highlights the importance of glucose and its influx in the metabolism of HNSCC cells [8]. In contrast, when SLC2A3 is elevated, the potential enhanced glucose influx is not enough to induce a radioresistant phenotype if the SDHA levels are low. Recently, Olszewski et al. [22] demonstrated that both reduced glucose entrance into the cell and low TCA enzyme expression are needed to reduce tumor growth. They showed that inhibiting class I glucose transporters was effective in inhibiting tumor growth in patient-derived xenograft models of SDHA-deficient tumors.

Analysis of SLC2A3 and SDHA transcriptional expression in patients with HNSCC from the TCGA dataset revealed trends consistent with those observed in our patients. Specifically, higher transcriptional expression of SLC2A3 and SDHA was associated with an increased likelihood of having an active tumor at the last contact or time of death. However, it is important to note that the characteristics of patients in the TCGA dataset differ significantly from those in our study, with a higher proportion of oral cavity carcinomas and a predominant use of surgical treatment.

There are several limitations in our study that need to be considered when analyzing the results. This is a retrospective study in which a limited number of genes related to glucose metabolism were studied. Moreover, we performed analysis only of the transcriptional expression, without providing information on the signaling pathways activated, the local concentration of substrates, or the existence of post-translational regulatory mechanisms. The limited number of patients with oral cavity carcinomas in this study may affect the generalizability of our findings to this specific sub-group. Additionally, while it would have been valuable to assess the expression of SLC2A and SDHA in recurrent tumors, we were unable to perform this analysis due to the lack of sufficient samples.

External validation studies are necessary before SLC2A3/SDHA transcriptional expression can be considered as a reliable biomarker for response to RT or CRT in patients with HNSCC. If confirmed in future studies, the predictive role of SLC2A3 and SDHA expression could contribute to more personalized treatment strategies, particularly in identifying patients at higher risk of local recurrence following radiotherapy. Furthermore, these findings may provide a foundation for future research into targeted therapies for patients with elevated SLC2A3 and SDHA expression.

## 4. Materials and Methods

### 4.1. Patients

The present study was performed retrospectively from biopsies obtained from the primary location of the tumor prior to any type of oncologic treatment in 120 patients with histologically confirmed squamous cell carcinomas located in the oral cavity, oropharynx, hypopharynx, or larynx treated with RT or CRT during the period 2008–2016. The patients included in the present study were diagnosed and treated in a different center than the one in which the initial determinations of the relationship between the succinate pathway and prognosis in patients with HNSCC were carried out [9]. Clinical information was obtained from a database that prospectively collects data related to the clinical characteristics, treatment, and follow-up of all patients with malignant head and neck tumors treated at our center. All patients included in the study were evaluated by a multidisciplinary tumor board, who proposed treatment with RT or CRT according to the institutional treatment protocols. In general, treatment was RT alone for patients with early-stage tumors (stages I–II) and CRT for advanced-stage tumors (stages III–IV), depending on the clinical characteristics of the patient. Table 4 shows the characteristics of the patients included in the study.

Given the interaction between tobacco and alcohol consumption, a new combined variable of toxics consumption was created with 3 categories: no consumption; moderate consumption (<20 cigarettes/day and/or <80 g alcohol/day); and severe consumption (≥20 cigarettes/day or ≥80 g alcohol/day). For patients with oropharyngeal carcinomas, information regarding the HPV status of the tumor was available for 42 of the patients included in the study. The presence of viral DNA was determined with RT-PCR with the SPF-10 PCR/DEIA/LiPA25 system until 2012 and with the PCR/CLART HPV2 system thereafter. For all positive HPV-DNA samples, immunohistochemical expression of p16INK4a was evaluated, considering as positive those specimens with intense and diffuse staining of more than 70% of the tumor tissue. HPV-related tumors (HPV-positive) were considered those with the presence of viral DNA together with immunopositivity to p16INK4a. Twelve oropharyngeal tumors (28.6%) were considered HPV-positive.

RT treatment given was 70 Gy at the primary tumor and morphologically and/or metabolically positive lymph nodes and 50 Gy in the lymph node areas at risk of microscopic disease, according to international consensus guidelines. CT treatment consisted in the administration of two to three cycles of cisplatin at a dose of 100 mg/m^2^ every 21 days (n = 59) or carboplatin administered weekly at a dose of 1.5 AUC (n = 7), administered concomitantly with RT.

All patients included in the study had a follow-up period of more than 3 years. The mean follow-up period of the patients was 5.2 years (standard deviation of 3.7 years).

### 4.2. Transcriptional Analysis

The biopsy samples obtained from each patient were immediately enclosed in RNA-later (Quiagen GmbH, Hilden, Germany) to prevent mRNA degradation and stored at −80 °C until processing. Total RNA was extracted using Trizol (Invitrogen, Carlsbad, CA, USA) according to the manufacturer’s instructions. The cDNA was obtained by reverse transcription of 1 µg RNA with High-Capacity cDNA Archive Kit (Applied Biosystems, Foster City, CA, USA), and transcriptional expression of SUCNR1, SLC2A3, SLC16A3, SDHA, and Beta-actin as endogenous control were assessed by RT-PCR on an ABI Prism 7000 using pre-designed validated assays (TaqMan Gene Expression Assays; Applied Biosystems).

### 4.3. External Validation Study: The Cancer Genome Atlas Database

We conducted an external validation study to assess the prognostic significance of SLC2A3 and SDHA transcriptional expression using data from The Cancer Genome Atlas (TCGA) [24]. Patients with available tumor status information were included in the analysis. Tumor status was defined as either tumor-free or with an active tumor at the last contact or time of death. Among the 473 patients with tumor status data, 334 (70.6%) were tumor-free, while 139 (29.4%) had active disease. The characteristics of these patients, stratified by tumor status, are presented in Appendix A of the Appendix A.

### 4.4. Statistical Analysis

We compared the transcriptional expression levels of SUCNR1, SLC2A3, SLC16A3, and SDHA according to sex, toxics consumption, location of the primary tumor, clinical local (cT) and regional (cN) extension of the tumor, and local control of the tumor after treatment with RT or CRT. The distribution of the transcriptional expression of the genes analyzed did not meet the criteria of normality, so we used the non-parametric Mann–Whitney U or Kruskal–Wallis tests in the comparisons of the transcriptional expression values.

The relationship between the transcriptional expression for each of the genes analyzed and local disease control after treatment was assessed with a recursive partitioning analysis (RPA) using the classification and regression tree model. If a relationship between local disease control and transcriptional expression was present, the RPA identified the transcriptional expression cut-off value with the highest prognostic capacity. Subsequently, genes that were associated with local disease control when analyzed individually were then included jointly in another RPA, considering the local control as the dependent variable. Local recurrence-free survival analysis was performed according to the categories obtained with the RPA using the Kaplan–Meier method. Differences between survival curves were analyzed with the log-rank test. A multivariable analysis was carried out considering local recurrence-free survival as the dependent variable, including as one of the independent variables the categories derived from the RPA.

The transcriptional expression values of SLC2A3 and SDHA in TCGA patients did not follow a normal distribution; therefore, we used the non-parametric Mann–Whitney U tests in the comparison of the transcriptional expression values according to tumor status. Subsequently, we performed an RPA, using SLC2A3 and SDHA expression values as predictors and tumor status as the dependent variable.

## 5. Conclusions

SLC2A3/SDHA transcriptional expression was significantly associated with local control in HNSCC patients treated with RT or CRT. Patients with a high transcriptional expression of both SLC2A3 and SDHA had a significantly higher risk of local recurrence after treatment with RT or CRT.

## Figures and Tables

**Figure 1 ijms-26-02451-f001:**
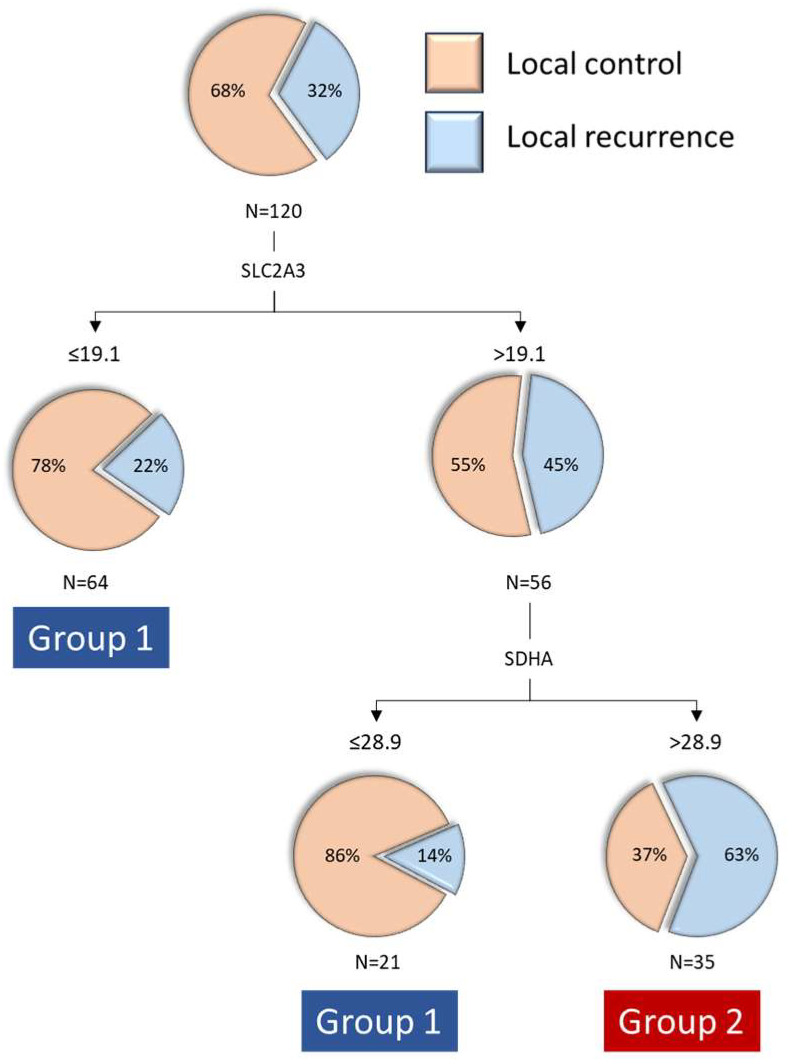
Classification and regression tree according to the transcriptional expression values of SLC2A3 and SDHA considering the local control after treatment with radiotherapy or chemoradiotherapy as the dependent variable.

**Figure 2 ijms-26-02451-f002:**
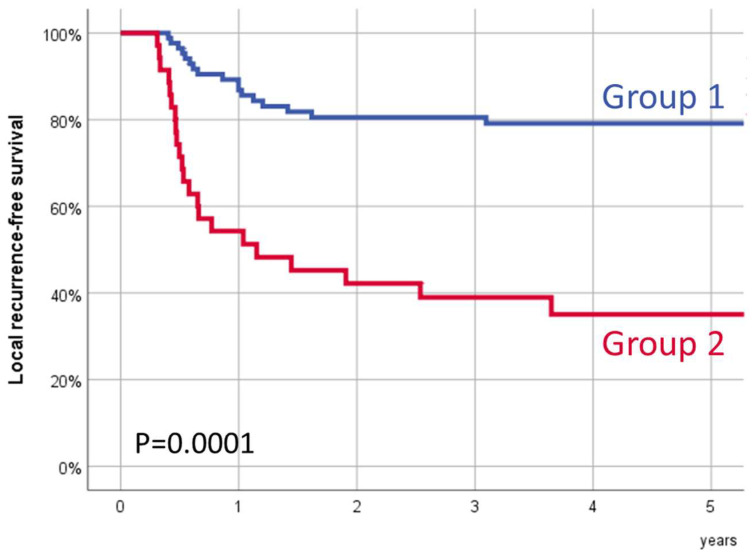
Local recurrence-free survival after treatment with radiotherapy or chemoradiotherapy according to SLC2A3 and SDHA transcriptional expression. Group 1, patients with low SLC2A3 expression or patients with high SLC2A3 and low SDHA expression; Group 2, patients with high SLC2A3 and high SDHA expression.

**Table 1 ijms-26-02451-t001:** Distribution of patients according to the cut-off points obtained with the classification and regression tree and 5-year local recurrence-free survival (LRFS) for each of the categories obtained with these cut-off points.

	Cut-Off	N	5–Year LRFS (95% CI)	*p*
SLC2A3	Low (≤19.16)	64	77.2% (66.6–87.8%)	0.005
	High (>19.16)	56	54.0% (40.7–67.3%)	
SLAC16A3	Low (≤40.83)	50	77.0% (65.0–89.0%)	0.025
	High (>40.83)	70	58.2% (46.2–70.2%)	
SDHA	Low (≤29.13)	51	79.5% (68.1–90.9%)	0.009
	High (>29.13)	69	56.5% (44.5–68.5%)	

**Table 2 ijms-26-02451-t002:** Results of the multivariable analysis considering local recurrence-free survival as the dependent variable.

		HR (CI95%)	*p*
Age		1.02 (0.98–1.06)	0.234
Sex	Male	1	
	Female	0.74 (0.25–2.22)	0.602
Toxics consumption	No	1	
	Moderate	0.44 (0.07–2.57)	0.367
	Severe	0.45 (0.10–1.99)	0.296
Location	Oral cavity	1	
	Oropharynx	0.84 (0.12–5.86)	0.865
	Hypopharynx	2.14 (0.25–17.99)	0.481
	Larynx	1.50 (0.19–11.33)	0.693
Local extension	cT1-T2	1	
	cT3-T4	2.29 (0.94–5.58)	0.066
Regional extension	cN0	1	
	cN+	1.04 (0.38–2.88)	0.929
Histological grade	Well differentiated	1	
	Moderately differentiated	2.99 (0.47–19.01)	0.244
	Poorly differentiated	0.89 (0.08–8.95)	0.925
Treatment	Radiotherapy	1	
	Chemoradiotherapy	2.20 (0.56–8.55)	0.253
SLC2A3-SDHA	Group 1	1	
	Group 2	4.24 (2.07–8.69)	0.0001

**Table 3 ijms-26-02451-t003:** Five-year local recurrence-free survival (LRFS) according to SLC2A3 and SDHA transcriptional expression.

Transcriptional SLC2A3/SDHA Expression	5-Year LRFS (95% CI)
High SLC2A3/low SDHA (n = 21)	85.2% (69.7–100%)
High SLC2A3/high SDHA (n = 35)	35.1% (18.6–51.6%)
Low SLC2A3/low SDHA (n = 30)	75.5% (59.6–91.4%)
Low SLC2A3/high SDHA (n = 34)	78.6% (64.5–92.7%)

**Table 4 ijms-26-02451-t004:** Characteristics of the patients included in the study.

		N (%)
Age	Mean 62.59 years (rank 38.1–98.7 years)	
Sex	Male	106 (88.3%)
	Female	14 (11.7%)
Toxics consumption	No	13 (10.8%)
	Moderate	17 (14.2%)
	Severe	90 (75.0%)
Location	Oral cavity	6 (5.0%)
	Oropharynx	50 (41.7%)
	Hypopharynx	15 (12.5%)
	Larynx	49 (40.8%)
Local extension	cT1	23 (19.2%)
	cT2	47 (39.2%)
	cT3	35 (29.2%)
	cT4	15 (12.5%)
Regional extension	cN0	71 (59.2%)
	cN1	14 (11.7%)
	cN2	33 (27.5%)
	cN3	2 (1.7%)
Stage	I	22 (18.3%)
	II	26 (21.7%)
	III	28 (23.3%)
	IV	44 (36.7%)
Histological grade	Well differentiated	11 (9.2%)
	Moderately differentiated	97 (80.8%)
	Poorly differentiated	12 (10.0%)
Treatment	Radiotherapy	54 (45.0%)
	Chemoradiotherapy	66 (55.0%)

## Data Availability

Data are available from the corresponding author upon reasonable request.

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
