# Peer review of "Transcriptional Expression of SLC2A3 and SDHA Predicts the Risk of Local Tumor Recurrence in Patients with Head and Neck Squamous Cell Carcinomas Treated Primarily with Radiotherapy or Chemoradiotherapy"

_ijms, 2025, doi:10.3390/ijms26062451_

Round 1

Reviewer 1 Report

Comments and Suggestions for Authors

In this article the authors investigated probable predictive markers for recurrence in HNSCC using 120 patients. They found SLC2A3 and SDHA to be markers to predict local recurrence with data leading to regional recurrence and distant metastasis. My comments are as follows:

  1. For supplementary figure S1, the legend is not clear. What does local recurrence mean, is it transcript from recurred tumor or primary tumor tumor which recurred?
  2. How the cut-off or high low of each gene was determined. Need to elaborate
  3. Statistical significance of fig 1 needed
  4. The authors should validate their finding using g other cohorts like the TCGA and how well the SLC2A3 and SDHA predict recurrence. It can be any recurrence. But validation will strengthen the point and help the study be basis for future clinical research
  5. It will be great if the level of SLC2A3 and SDHA are checked in recurred tumors. Is there an increase? Does cells high in SLC2A3 and SDHA only recur. This finding will help in future research where those cells having SLC2A3 and SDHA can be targeted. A similar study was carried out for another protein EpCAM, for reference look at PMID: 33490064

Author Response

In this article the authors investigated probable predictive markers for recurrence in HNSCC using 120 patients. They found SLC2A3 and SDHA to be markers to predict local recurrence with data leading to regional recurrence and distant metastasis. My comments are as follows:

Thank you very much for taking the time to review this manuscript. Please find the detailed responses below and the corresponding revisions highlighted in the re-submitted files.

1.    For supplementary figure S1, the legend is not clear. What does local recurrence mean, is it transcript from recurred tumor or primary tumor which recurred?

Response 1. Figure S1 illustrates the pretreatment transcriptional expression of the tumor based on local control outcomes following radiotherapy. In the reviewed manuscript, we have updated the figure legend and included the number of patients in each analyzed group.

2.    How the cut-off or high low of each gene was determined. Need to elaborate

Response 2. The cut-off points for local tumor control following radiotherapy were determined using Recursive Partitioning Analysis (RPA). If a relationship existed between transcriptional expression and local control, RPA identified the threshold with the highest discriminatory power. This concept has been incorporated into the statistical analysis section of the revised manuscript.

“If a relationship between local disease control and transcriptional expression was present, the RPA identified the transcriptional expression cut-off value with the highest prognostic capacity.”

3.    Statistical significance of fig 1 needed.

Response 3. Figure 1 shows the classification tree obtained from the RPA analysis. The survival data obtained from this classification are shown in Figure 2, which shows the level of statistical significance.

4.    The authors should validate their finding using g other cohorts like the TCGA and how well the SLC2A3 and SDHA predict recurrence. It can be any recurrence. But validation will strengthen the point and help the study be basis for future clinical research

Response 4. In accordance with the reviewer's suggestion, we have performed a validation of the prognostic ability of SLC2A3 and SDHA transcriptional expression using TCGA data. Although this is a population of HNSCC patients with very different characteristics from our cohort of patients, with a majority of tumors located in the oral cavity and treated with surgery, and the data included in TCGA does not allow specific differentiation of local disease control, we were able to verify how the expression of SLC2A3 and SDHA was significantly related to tumor status. In the revised version of the manuscript, we have included the results obtained in this validation study.

5.    It will be great if the level of SLC2A3 and SDHA are checked in recurred tumors. Is there an increase? Does cells high in SLC2A3 and SDHA only recur. This finding will help in future research where those cells having SLC2A3 and SDHA can be targeted. A similar study was carried out for another protein EpCAM, for reference look at PMID: 33490064

Response 5. We agree with the reviewer that determination of the transcriptional expression of SLC2A3 and SDHA in tumor recurrence would be interesting. Unfortunately, we do not have samples of tumor recurrences in which to perform such determination.

Reviewer 2 Report

Comments and Suggestions for Authors

1. The level of SLC2A3 and SDHA expression appears associated with prognosis of cancer in other manuscripts, so this manuscript is lack od novelty to some extent.

2. The title and content of the article do not seem to completely match. The title is " Transcriptional expression of SLC2A3 and SDHA predicts the risk of local tumor recurrence in patients with head and neck squamous cell carcinomas treated with radiotherapy", but in actual research results, similar results have also been shown in chemotherapy.

3. In terms of patient gender selection, there are 106 males and 14 females, with a significant disparity. Can you explain why this is the case?

4. Why did the author consider transcriptional expressions of SLC2A3 and SDHA, not protein expressions as biomarkers for detecting head and neck cancer?

5. Have the transcriptional expressions of SLC2A3 and SDHA been compared with other highly recognized indicators for predicting head and neck cancer?

Author Response

Thank you very much for taking the time to review this manuscript. Please find the detailed responses below and the corresponding revisions highlighted in the re-submitted files.

1.    The level of SLC2A3 and SDHA expression appears associated with prognosis of cancer in other manuscripts, so this manuscript is lack od novelty to some extent.
Response 1. We agree with the reviewer that SLC2A3 expression has been related to the prognosis in patients with HNSCC. However, as discussed, this relationship has been established through immunohistochemical analysis (Ayala et al. ref 11; Baer et al.ref 13), or through transcriptional expression assessment (Estilo et al. ref 12) in patients with oral cavity carcinomas treated with surgery. To our knowledge, our study is the first to evaluate the predictive capacity of SLC2A3 transcriptional expression in patients treated with radiotherapy.

2.    The title and content of the article do not seem to completely match. The title is " Transcriptional expression of SLC2A3 and SDHA predicts the risk of local tumor recurrence in patients with head and neck squamous cell carcinomas treated with radiotherapy", but in actual research results, similar results have also been shown in chemotherapy.
Response 2. In agreement with the reviewer's suggestion, we have modified the title of the study to include the term chemo-radiotherapy.

3.    In terms of patient gender selection, there are 106 males and 14 females, with a significant disparity. Can you explain why this is the case?
Response 3. In our epidemiological context, where the majority of HNSCC cases are related to tobacco and alcohol use, the incidence of HNSCC is notably higher in in male patients than in female patients. ( León X, López M, García J, Montserrat JR, Gras JR, Kolanczak KA, Quer M. Epidemiologic characteristics of squamous head and neck carcinoma patients. Results of a hospital register. Acta Otorrinolaringol Esp (Engl Ed). 2019 Sep-Oct;70(5):272-278.. doi: 10.1016/j.otorri.2018.05.006). 
We do not believe it is necessary to include a comment on this subject in the manuscript. However, if needed according to the editor or reviewer, we could include it to justify the observed sex-based differences in out cohort.

4. Why did the author consider transcriptional expressions of SLC2A3 and SDHA, not protein expressions as biomarkers for detecting head and neck cancer?
Response 4. This study aims to analyze whether the transcriptional expression of SLC2A3 and SDHA is associated with local control following radiotherapy. Other studies have explored the prognostic relationship between the immunohistochemical expression of these biomarkers in patients with HNSCC. 
5. Have the transcriptional expressions of SLC2A3 and SDHA been compared with other highly recognized indicators for predicting head and neck cancer?
Response 5. We agree with the reviewer's interest in carrying out this type of analysis. However, such comparisons are outside the scope of the current study.

Round 2

Reviewer 1 Report

Comments and Suggestions for Authors

The authors have successfully answered my queries.

Author Response

We appreciate the revision carried out by the reviewer of our manuscript.

Reviewer 2 Report

Comments and Suggestions for Authors

The data volume and novelty of this article still need to be improved.

Author Response

We respectfully disagree with the reviewer's assessment regarding the originality of our manuscript. While previous studies have indeed investigated the prognostic role of SLC2A expression in patients with head and neck carcinomas, these studies have been limited to specific tumor sites, such as the oral cavity (Refs. 11, 12) or the larynx (Refs. 13, 14). Additionally, these studies predominantly focused on cohorts treated primarily with surgery (Refs. 11, 12, 13, 14) and assessed SLC2A3 expression through immunohistochemistry (Refs. 11, 13). To the best of our knowledge, no studies have yet explored the predictive potential of SLC2A3 transcriptional expression in relation to treatment response in patients with HNSCC from different sites treated with radiotherapy or chemo-radiotherapy.

Furthermore, there is a notable paucity of data regarding the predictive value of SDHA expression in HNSCC. As referenced in our manuscript, our group previously conducted a preliminary study that identified a correlation between SDHA transcriptional expression and loco-regional control in a sub-cohort of patients treated with radiotherapy or chemo-radiotherapy. In the present study, we expand on these findings by validating the predictive capacity of SDHA transcriptional expression, examining it in combination with SLC2A3.